# Is Female a More Pro-Environmental Gender? Evidence from China

**DOI:** 10.3390/ijerph19138002

**Published:** 2022-06-29

**Authors:** Yong Li, Bairong Wang, Orachorn Saechang

**Affiliations:** 1School of Marxism, Shanghai Maritime University, Shanghai 201306, China; liyong@shmtu.edu.cn; 2School of Economics and Management, Shanghai Maritime University, Shanghai 201306, China; 3Faculty of Political Science and Public Administration, Chiang Mai University, Chiang Mai 50200, Thailand; orachorn.s@cmu.ac.th

**Keywords:** gender, female, pro-environmental psychology, pro-environmental behavior, China

## Abstract

The purpose of this study is to determine whether there are gender differences in people’s pro-environmental psychology and behaviors in China. An online survey was conducted with the snowball sampling technique, and a sample of 532 Chinese respondents was obtained for the research. This study finds that gender does affect green psychology and behaviors, with females reporting a higher level of environmentalism in China. Specifically, females are more concerned with environmental problems, more supportive of plastic ban policies, more positive towards reducing plastics (reduce), and have stronger intention to bring a reusable bag for shopping (reuse and recycle). Moreover, females use fewer disposable toiletries when checking in a hotel and require less disposable tableware when ordering takeout. This study contributes to the current literature by identifying the relationship between gender and environmentalism in China. Implications for anti-plastic policy design and environmental management are also presented.

## 1. Introduction

Environmentalism has gained more and more attention given the increasingly severe environmental problems in recent years. One possible way to expand environmentalism is to encourage green psychology and behaviors at the individual level. Efforts are extensive in examining the relationship between gender and pro-environmental psychology and behaviors [1,2,3] Research in some countries, such as Argentina, Canada, Spain, and the United States, finds that, due to socialization and social role differences, compared with males, females have a more favorable attitude towards environment [2]. Females are usually socialized to be the caregiver, and males are usually socialized to be the breadwinner [4,5]. Similarly, some environmental sociology literature shows that females express greater levels of environmental concern than males [4,6,7]. These gender differences also exist in pro-environmental behaviors, and females are found to be more active in pro-environmental actions [1,8,9,10,11].

It is reported that economic development causes environmental problems [12,13,14]. Developed countries face environmental issues earlier and more seriously. Accordingly, environmental research is far more extensive in developed countries [15]. China is a developing country. As for the specific topic of gender and environmentalism, there is limited research in China. To fill the research gap, this study aims to examine gender differences in Chinese people’s pro-environmental psychology and behaviors.

The innovation of this study lies in a deeper dive into gender distinctions in individuals’ environmental psychology regarding plastic management from the perspectives of 3R, i.e., reduce, reuse, and recycle [16]. The extensive use of plastic products has aroused worldwide concerns [17]. To solve the plastic crisis, many countries have launched different plastic ban policies [17,18,19]. For example, to improve the plastic-reducing effects, on the basis of the 2008 plastic ban policies, the Chinese government issued tougher plastic ban policies in 2020 [20]. However, people respond differently to the new tougher policies, which are also barely studied. Thus, we are driven to examine gender differences in responding to the new tougher policies.

## 2. Gender and Environmentalism

### 2.1. Gender and Pro-Environmental Psychology

Socialization and the resulting different gender roles are a widely utilized theoretical approach to examine the relationship between gender differences and environmental attitudes [1,21]. Research has found that females are socialized to be more caring, altruistic, cooperative, and helpful, while males are socialized to be more independent and competitive [2,5,22]. As a result of different expectations for men and women, environmental concern may vary by gender. In general, females are more inclined to foster bonds with nature and possess a higher level of concern for the environment than males [23,24,25]. For instance, females tend to be more concerned about environmental pollution [4]. Likewise, Tikka, Kuitunen, and Tynys [26] find that female students show more environmental responsibility than male students.

However, some studies find no significant differences between females and males regarding their pro-environmental psychology. For instance, Mohai [27] argues that no firm conclusions can be drawn regarding the influence of gender on various environmental issues. Similarly, Arcury and Christianson [28] find no significant differences between males and females on environmental concern. Additionally, Mostafa [29] points out that males express higher levels of environmental concern in Egypt. Males are also found to be a greener gender in India [30]. Thus, this study is motivated to examine whether gender differences in pro-environmental psychology could be found in the Chinese context.

**Hypothesis** **1** **(H1).**
*Females show more environmental concern and stronger pro-environmental attitudes than males.*


### 2.2. Gender and Pro-Environmental Behaviors

When predicting pro-environmental behaviors, gender is a powerful influential factor [9]. Sahin, Ertepinar, and Teksoz [31] find that female students take part in more pro-environmental activities and adopt a more sustainable lifestyle than males. Females are found to be more active in conservation behavior than males [11]. Likewise, Wang and Li [17] suggest that female consumers significantly use fewer plastic bags and more reusable bags than male consumers do for shopping.

However, Davidson and Freudenburg [21] suggest that gender differences in environmentalism are not universal. Hunter, Hatch, and Johnson [32] find that females participate more in private environmental actions, while there are no consistent gender differences in publicly oriented activities. Blocker and Eckberg [4] find no significant differences between females and males in recycling. Berenguer, Corraliza, and Martin [33] also fail to find significant gender differences in anti-pollution behaviors. On the contrary, some research suggests that males engage in more environmental actions than females [1,34,35,36]. As discussed above, existing studies on the relationship between gender and green behaviors have yielded mixed results. Therefore, this study aims to investigate whether gender differences in pro-environmental behaviors exist in China.

**Hypothesis** **2** **(H2).**
*Females behave more pro-environmental than males.*


## 3. Materials and Methods

### 3.1. Design and Sample

Based on the findings of a similar study done by Wang and Li [17], which proposes that females would have a more positive green psychology and conduct more positive green actions than males in China, this study firstly examines gender differences in environmental psychology, including environmental concern, support for plastic ban policies, attitudes towards reducing plastics, and intention to bring a reusable bag for shopping. Second, this study investigates gender differences in pro-environmental behaviors, including green travel behavior, green ordering takeout behavior, and water conservation behavior. The research model of this study is displayed in Figure 1.

This study conducted a survey in the latter half of 2021 to evaluate respondents’ green psychology and behaviors in China. Due to the impact of COVID-19, for safety concerns, we distributed questionnaires online to the general public in China using a snowballing technique [37]. Recruitment of respondents was completed through social media platforms utilizing preexisting social and personal contacts. The questionnaire was pilot tested on 25 respondents to revise the wording of the survey items so that the statements were appropriate. The details of this survey are displayed in Appendix A. A total of 534 questionnaires were obtained, and 532 of them were valid. This study passes the Kendall Tau test, indicating that non-response bias does not exist [38].

### 3.2. Dependent Variables

Four kinds of pro-environmental psychology and three pro-environmental behaviors are measured in the survey. In terms of pro-environmental psychology variables, environmental concern was measured by 13 items adapted from Minton and Rose [39]. A 5-point Likert scale was used for evaluation, in which 1 means “strongly disagree” and 5 means “strongly agree”. The sampling question is “Public schools should require all students to take a course dealing with the environment and conservation problems”. The other three pro-environmental psychology variables include support for plastic ban policies, attitudes towards reducing plastics (reduce), and intention to bring a reusable bag for shopping (reuse and recycle), respectively. The intention to bring a reusable bag for shopping was measured by the scale developed by Wang and Li [40]. The sampling question is “I am willing to bring a reusable bag for shopping”. A 5-point Likert scale was used for evaluation, in which 1 means “strongly disagree” and 5 means “strongly agree”. Regarding pro-environmental behavior variables, green travel behavior, green ordering takeout behavior, and water conservation behavior were measured in this study. The answers were coded as 1 for yes and 0 for no. Details on response patterns for the seven dependent variables are summarized in Table 1.

### 3.3. Independent Variables

Summary information on response patterns for independent variables is shown in Table 2. The key independent variable was respondents’ gender, with 55.6% females and 44.4% males in this study. Five control variables were also incorporated into the analysis, namely age, education, marital status, monthly income, and environmental knowledge. Participants younger than 30 made up 42.7% of the sample, followed by 30–39-year-olds (37.8%). The majority of the sample had a Bachelor’s degree, accounting for 47.9%. About half of the participants (52.6%) were married. As for monthly income, 32.3% of the participants earned less than RMB 5000, and 22.2% of the participants earned between RMB 5000–7999. Previous studies demonstrated that environmental knowledge was an essential prerequisite for environmentalism [29,41]. In this study, environmental knowledge was measured by the question “How often do you get access to environmental knowledge, such as watching documentaries, TV programs, or short videos related to environmental protection?” Over half of the participants (57.1%) sometimes gained access to environmental knowledge, and only 17.3% of the participants could usually or always gain access to environmental knowledge.

### 3.4. Data Analysis

This study utilized linear regression and binary logistic regression models to test whether there were significant associations between gender and four indicators of pro-environmental psychology, as well as between gender and three indicators of pro-environmental behaviors, respectively.

## 4. Results

Table 3 reports the descriptive statistics of the analyzed variables. The OLS regression results for individuals’ pro-environmental psychology and binary logistic regression results for individuals’ pro-environmental behaviors are presented in Table 4 and Table 5, respectively.

Based on the results of Table 4 and Table 5, gender differences are significant in pro-environmental psychology and behaviors in the present study. As shown in Model 1 of Table 4, females express significantly (*β* = 0.158, *p* < 0.01) greater environmental concern than males, echoing the study of Casey and Scott [23], which also finds female to be associated with higher levels of environmental concern. The results from Models 2–4 show that the effects of gender on environmental psychology regarding plastic management are strong and consistent, including support for plastic ban policies, attitudes towards reducing plastics (reduce), and intention to bring a reusable bag for shopping (reuse and recycle). Specifically, females are more likely to support China’s 2020 plastic ban policies (*β* = 0.271, *p* < 0.001). Moreover, compared with males, females tend to have more positive attitudes towards reducing plastics (*β* = 0.218, *p* < 0.05) and show stronger intention to bring a reusable bag for shopping (*β* = 0.300, *p* < 0.001). Hence, it is suggested that females could exert more positive influence on future plastic reduction activities. H1 is supported. Additionally, for all these four types of green psychology, environmental knowledge is a powerful influential factor (see Models 1–4).

The analysis results on the prediction of the probability of individuals’ pro-environmental behaviors within the binary logistic regression models are presented in Table 5. Females are more likely to conduct green travel behavior and green ordering takeout behavior (see Models 5–6). To be specific, the likelihood of females bringing their own toiletries when checking in the hotel is 1.518-times higher than males (see Model 5). Moreover, the likelihood of females requiring non-disposable -tableware when ordering takeout is 1.674-times higher than the likelihood of males (see Model 6). However, gender does not significantly influence individuals’ water conservation behavior (see Model 7). That is to say, the effects of gender vary by type of pro-environmental behaviors. A possible explanation could be that the pro-environmental behaviors of green travel and green ordering takeout are occasional, but water conservation is more likely to be a habit or lifestyle. In terms of a sustainable lifestyle, this study finds that females do not necessarily do better than males. Hence, the antecedents of a sustainable lifestyle or green habits still need further research. H2 is partially supported. Additionally, environmental knowledge is a persistently influential factor for all three kinds of green behaviors (see Models 5–7).

## 5. Discussion

This study examines individuals’ gender differences in pro-environmental psychology and behaviors in China. In terms of the relationship between gender and environmentalism, a clear picture has developed. Females are usually a more pro-environmental gender in China as they are generally more concerned about the environment and conduct more green behaviors. Based on the research findings, several valuable implications are presented as follows.

First, this study identifies similar gender differences in environmental concern in China, which strengthens and echoes the socialization theory that indicates females have higher environmental concern [2,5]. Moreover, the research findings show that females tend to be more supportive of plastic ban policies, show more positive attitudes towards reducing plastics (reduce), and exhibit a stronger intention to bring a reusable bag for shopping (reuse and recycle). To raise the general public’s level of environmental concern, males may be more targeted in future environmental education. This finding also has important insights for anti-plastic policy popularization and implementation. Specifically, future publicity of plastic ban policies could rely more on females, who could be better informed and aware of these policies.

Second, existing studies show that higher levels of environmental concern could translate into pro-environmental behaviors [1,42]. This study partially verifies this view by revealing that females engage more in green travel and green ordering takeout actions, while regarding water conservation, no significant gender differences are found in the present study. Based on the research results, we hesitate to suggest that female is a greener gender in all environmental issues.

Third, this research confirms that environmental knowledge has a significantly positive influence on both individuals’ green psychology and behaviors, which is in line with the findings of Mostafa [29] and Paço and Lavrador [41]. Environmental knowledge could be regarded as a necessary prerequisite for individuals’ environmentalism. As suggested in this study, environmental knowledge received through the mass media, such as watching documentaries, TV programs, or short videos related to environmental protection, is beneficial in generating more eco-responsible individuals.

## 6. Conclusions

This study finds that gender does affect green psychology and behaviors, with females reporting a higher level of environmentalism in China. Specifically, females are more concerned with environmental problems, more supportive of plastic ban policies, more positive attitudes towards reducing plastics (reduce), and have stronger intention to bring a reusable bag for shopping (reuse and recycle). Moreover, females use fewer disposable toiletries when checking in a hotel and require less disposable tableware when ordering takeout. This study contributes to the current literature by identifying the relationship between gender and environmentalism in China. It is also observed that individuals with higher environmental knowledge tend to be more environmentally friendly.

## 7. Limitations and Future Research

First, as this study relied on self-report measurement, social desirability may affect the accuracy of the research findings. Second, this study discussed four indicators of pro-environmental psychology and three indicators of pro-environmental behaviors, which are far from comprehensive for all kinds of pro-environmental issues. Future research could involve a wider range of green psychological variables and behaviors. Third, mediating or moderating factors that may influence the gender–environmentalism relationship deserve more future research efforts.

## Figures and Tables

**Figure 1 ijerph-19-08002-f001:**
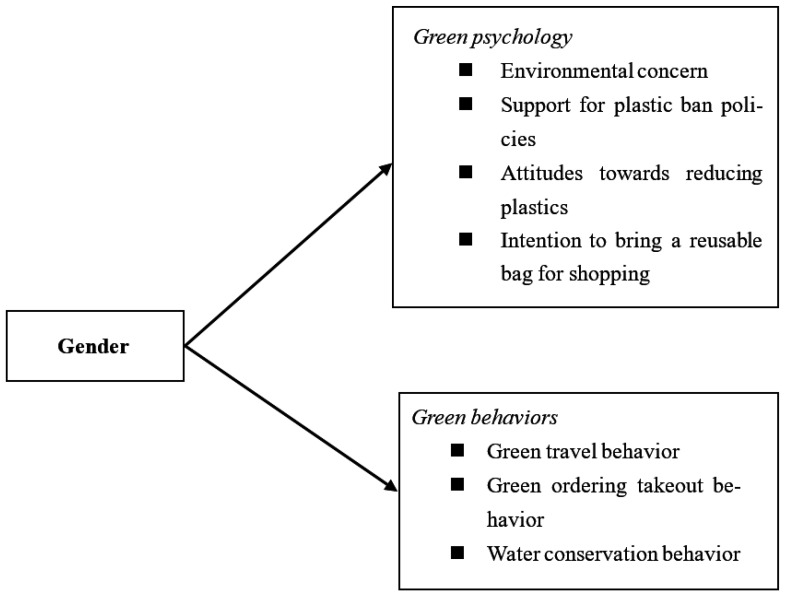
Research model examining gender differences in environmental psychology and behaviors.

**Table 1 ijerph-19-08002-t001:** Summary of response to dependent variables.

Variable	Question Item or Source	Category	N	Percentage (%)
** *Green psychology (Variables 1–4)* **				
1. Environmental concern	Minton and Rose (1997)	Strongly disagree/disagree	7	1.3
		Neutral	6	1.1
		Agree/strongly agree	519	97.6
2. Support for plastic ban policies	To what extent do you support the 2020 plastic ban policies?	Strongly nonsupport/nonsupport	22	4.1
		Neutral	90	16.9
		Support/strongly support	420	79
3. Attitudes towards reducing plastics	How necessary do you think that we should limit the use of plastic bags?	Very unnecessary/unnecessary	53	10
		Neutral	106	29.9
		Necessary/very necessary	373	60.1
4. Intention to bring a reusable bag for shopping	Wang and Li (2022)	Strongly disagree/disagree	29	5.5
		Neutral	73	13.7
		Agree/strongly agree	430	80.8
** *Green behaviors (Variables 5–7)* **				
5. Green travel behavior	Do you usually choose disposable toiletries when checking in the hotel?	Yes	340	63.9
		No	192	36.1
6. Green ordering takeout behavior	Do you usually choose disposable tableware when you order takeout?	Yes	324	60.9
		No	208	39.1
7. Water conservation behavior	Do you have the habit of reusing water in your daily life, such as keeping the water for washing vegetables to flush the toilet?	Yes	285	53.6
		No	247	46.4

**Table 2 ijerph-19-08002-t002:** Summary of response patterns for independent variables.

Variable	Category	N	Percentage (%)
Gender	Female	296	55.6
	Male	236	44.4
	Total	532	100
Age	Less than 30	227	42.7
	30–39	201	37.8
	40–49	68	12.8
	50 and more	36	6.8
	Total	532	100
Education	High school or lower	78	14.7
	Bachelor’s or an equivalent	255	47.9
	Master’s degree	116	21.8
	Ph.D. Total	83 532	15.6 100
Marital status	Married	280	52.6
	Single	252	47.4
	Total	532	100
Monthly income	RMB 0–4999	172	32.3
	RMB 5000–7999	118	22.2
	RMB 8000–9999	80	15
	RMB 10,000–14,999	82	15.4
	RMB 15,000–19,999	40	7.5
	RMB 20,000 or more	40	7.5
	Total	532	100
Environmental knowledge	Never/rarely	136	25.6
	Sometimes	304	57.1
	Usually/always	92	17.3
	Total	532	100

**Table 3 ijerph-19-08002-t003:** Mean, standard deviation, and correlation matrix of all variables included in the analyses.

Variable	Mean	SD	1	2	3	4	5	6	7	8	9	10	11	12	13
1. Age	32.735	9.512	1												
2. Education	3.327	1.026	−0.248 **	1											
3. Marital status	0.526	0.450	0.539 **	−0.116 **	1										
4. Monthly income	2.662	1.593	0.151 **	0.276 **	0.198 **	1									
5. Environmental knowledge	2.868	0.923	0.019	−0.010	−0.017	−0.028	1								
6. Gender	0.556	0.497	−0.068	−0.062	0.039	−0.135 **	−0.013	1							
7. Environmental concern	4.172	0.591	0.036	0.039	−0.011	0.078	0.150 **	0.108 *	1						
8. Support for plastic ban policies	4.079	0.879	0.076	0.026	0.068	0.058	0.194 **	0.136 **	0.689 **	1					
9. Attitudes towards reducing plastics	3.900	1.039	0.068	0.020	0.047	−0.018	0.251 **	0.097 *	0.380 **	0.433 **	1				
10. Intention to bring a reusable bag for shopping	3.958	0.850	0.151 **	0.021	0.120 **	0.094 *	0.197 **	0.150 **	0.582 **	0.482 **	0.326 **	1			
11. Green travel behavior	0.361	0.481	0.140 **	−0.064	0.101 *	−0.020	0.192 **	0.088 *	0.060 **	0.120 **	0.140 **	0.237 **	1		
12. Green ordering takeout behavior	0.391	0.488	0.239 **	0.030	0.228 **	0.044	0.164 **	0.095 *	0.066	0.138 **	0.088 *	0.192 **	0.336 **	1	
13. Water conservation behavior	0.536	0.499	0.028	−0.115 **	−0.023	−0.139 **	0.170 **	0.018	0.105 *	0.092 *	0.216 **	0.204 **	0.142 **	0.028	1

Notes: N = 532. Standard errors given in parentheses. * *p* < 0.05, ** *p* < 0.01.

**Table 4 ijerph-19-08002-t004:** OLS regression results for predictors of individuals’ environmental psychology.

	Environmental Concern (Model 1)	Support for Plastic Ban Policies (Model 2)	Attitudes Towards Reducing Plastics (Model 3)	Intention to Bring a Reusable Bag for Shopping (Model 4)
Age	0.004 (0.003)	0.007 (0.005)	0.009 (0.006)	0.013 ** (0.005)
Education	0.019 (0.027)	0.036 (0.039)	0.060 (0.046)	0.042 (0.038)
Marital status ^a^	−0.080 (0.061)	0.033 (0.089)	0.033 (0.105)	0.048 (0.085)
Monthly income	0.035 * (0.017)	0.032 (0.025)	−0.019 (0.030)	0.044 (0.024)
Environmental knowledge	0.097 *** (0.027)	0.187 *** (0.040)	0.283 *** (0.047)	0.184 *** (0.038)
**Gender** ^b^	0.158 ** (0.051)	0.271 *** (0.076)	0.218 * (0.089)	0.300 *** (0.072)
Constant	3.546 *** (0.168)	2.945 *** (0.247)	2.515 *** (0.290)	2.563 *** (0.235)
Adjusted R^2^	0.038	0.060	0.071	0.089
F	4.516 ***	6.617 ***	7.714 ***	9.679 ***
VIF	1.002–1.528	1.002–1.528	1.002–1.528	1.002–1.528

Notes: N = 532. Standard errors in parentheses. * *p* < 0.05, ** *p* < 0.01, *** *p* < 0.001. Reference categories: ^a^ marital status = single, ^b^ gender = male.

**Table 5 ijerph-19-08002-t005:** Binary logistic regression analysis of predictors of individuals’ pro-environmental behaviors.

Variables	Green Travel Behavior (Model 5)		Green Ordering Takeout Behavior (Model 6)		Water Conservation Behavior (Model 7)	
	b	Exp(b)	b	Exp(b)	b	Exp(b)
Age	0.027 *	1.028	0.047 ***	1.048	0.008	1.008
Education	−0.026	0.975	0.253 *	1.288	−0.157	0.855
Marital status ^a^	0.173	1.189	0.611 **	1.843	−0.112	0.894
Monthly income	−0.035	0.966	−0.032	0.969	−0.149 *	0.862
Environmental knowledge	0.467 ***	1.595	0.414 ***	1.513	0.379 ***	1.460
**Gender** ^b^	0.417 *	1.518	0.515 **	1.674	0.014	1.014
Constant	−2.994 **	0.050	−4.600 ***	0.010	−0.243	0.785
−2 Log likelihood	658.934		645.746		720.278	
χ^2^(*df*)	36.853 (6)		66.265 (6)		29.479 (6)	
Cox and Snell R^2^	0.067		0.117		0.054	
Nagelkerke R^2^	0.092		0.159		0.072	

Notes: N = 532. * *p* < 0.05, ** *p* < 0.01, *** *p* < 0.001. Exp(b) is the factor change in the odds of the dependent variable due to a one-unit increase in the specific independent variable. Reference categories: ^a^ marital status = single, ^b^ gender = male.

## Data Availability

The data that support the findings of this study is available upon request to the corresponding author.

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
