# Peer review of "Is Female a More Pro-Environmental Gender? Evidence from China"

_ijerph, 2022, doi:10.3390/ijerph19138002_

Round 1
Reviewer 1 Report
You wrote: " research on this topic is scarce in China. " I thinkit could be explian why are scarce in China and not at the international leve.
Do you have hypothesisi or assumption to add to the 2. Gender and Environmentalism chapter
The survey was distirubuted online, how? in hte paper is not specified.
You put the green psicology some variables like: Intention to bring a reusable bag for shopping; are you sure that this is not a behaviour? check also the others
In table 1 you put together in the same column:
|
Question Item or Source |
but in my opinion is not clear why they are together
Discussion and coclusions are in the same chapter? or do you miss the conclusions?
Author Response
Thank you for your time and efforts. We really appreciate it. Please find the attachment for more response details.
Best

Reviewer 2 Report
while the topic is worthy of investigation, the research design and methodology discussion fail short in two critical issues:
(1) what about non-response bias: how was it treated, any clue on that? provide a Kendall Tau test for non response bias (https://www.researchgate.net/publication/265441450_Using_strength_of_opinion_to_test_for_nonresponse_bias_in_mail_surveys)
(2) questionnaire should be presented at the appendix, with detailed scales for each question
Author Response

(The authors gave the same response as above.)

Round 2
Reviewer 1 Report
Now the paper is improved.
Reviewer 2 Report
acce